# Research on the economic effects of housing support expenditures under the perspective of consumption heterogeneity: Evidence from China

Li Shang[1], Decai Tang[2], Xiaoling Zhang[3], Cunshu Li[1], Nan Pan[4], Chunfang Huang[1], Aijun Sun[1]*

1 School of Business, Jiangsu Second Normal University, Nanjing, China, 2 School of Management Science and Engineering, Nanjing University of Information Science and Technology, Nanjing, China, 3 School of Law and Business, Sanjiang University, Nanjing, China, 4 Human Resources Department, Jiangsu Second Normal University, Nanjing, China

* hasaj@163.com

## Abstract

What kind of impact does the government's housing support expenditure have on residents' consumption? This is a topic that deserves in-depth study and is of practical significance. This study constructs provincial equilibrium panel data based on China's guaranteed housing construction and financial expenditures on housing support data from 1999–2009 and 2000–2021. It applies the systematic GMM method to estimate the impact of government housing support expenditures on residents' consumption. The study found that whatever form of expenditure on housing support contributed to the total consumption of urban residents, while the impact on the consumption structure had different results. Based on the divisions of consumption structure, the results of the increase in government housing support expenditure on the consumption structure of urban residents are different. An examination of different forms of housing support reveals that the predominantly secure form of housing construction has a positive effect on all consumption structure divisions. Whereas the predominantly monetary subsidy form has a significant positive relationship with housing, necessity, and durability consumption expenditures, it has a weak or even negative relationship with non-housing, non-necessity, and non-durability consumption expenditures. The research in this paper makes up for the lack of current literature examining the economic effects of housing support from the perspective of consumption structure and provides a theoretical basis and policy reference for constructing a multi-level gradient housing support system.

## 1. Introduction

Consumption is an important factor in stimulating economic growth. Relative to exports and investment, the expansion of consumer demand as an important condition for improving the

**Funding:** The author(s) received no specific funding for this work.

**Competing interests:** The authors have declared that no competing interests exist.

quality of economic growth and enhancing the endogenous momentum of economic growth, which is of great significance for maintaining sound and rapid economic development. On the whole, China's consumption rate is lower than that of other countries at comparable levels of international development and even lower than the average level of developed countries. Housing support for residents is one of the livelihood issues that governments around the world are paying close attention to. Housing support is an important system led by the government, with the main purpose of solving the housing problems of low- and middle-income residents. Housing support expenditure is a major government expenditure, and the role of government functions and economic development is pivotal. A good support system can effectively solve the housing problem of residents, reduce the burden of housing, and release social consumption so that more people can consume to promote sustained and healthy socio-economic development. The question that arouses our research interest is whether there is a correlation between the government's housing support system and residents' consumption. Further, do different forms of housing support expenditure impact residents' consumption differently? What is the role of housing support expenditure on the consumption structure of the population?

This paper carries out validation and extension research on the issue of housing support expenditure. Firstly, it is concluded that the increase in government housing support expenditure is favorable to the increase in urban residents' total consumption expenditure. This study's outcome is consistent with the findings of scholars like Sebastian Gechert et al. (2021) [1]. Government expenditure on housing support is part of social support expenditure and part of government expenditure on people's livelihood. This part of the expenditure also has the attribute of government investment, such as the construction of guaranteed housing. The government's expansion of housing support expenditure will, on the one hand, increase the construction of guaranteed housing and, on the other hand, increase monetary subsidies. The government's direct or indirect subsidies to the beneficiary population can effectively reduce their expenditures on renting and purchasing housing, thus expanding the total consumption expenditures. Secondly, it is discovered that government housing support expenditure impacts the consumption structure of urban residents. In terms of consumption of necessities and non-necessities, the findings of this paper have similarities with those of Camacho-Rivera M et al. (2017) [2] et al. Camacho-Rivera M et al. (2017) [2] argued that the government's public housing program helps beneficiaries increase their consumption of necessities such as food and drink, and the conclusions of this study show that any increase in the Chinese government's expenditure on housing support increases urban residents' consumption of necessities. In terms of the consumption of durable goods and non-durable goods, the findings of this paper are consistent with the findings of Liaw C (2023) [3] et al. that the increase in housing support expenditure contributes to the increase in the consumption of durable goods by urban residents. This paper also found that two different forms of housing support expenditure seem to present different results in terms of their impact on the consumption structure of urban residents. The form of housing support expenditure based on the construction of guaranteed housing, regardless of the form of division, plays a role in promoting this consumption expenditure, while the form of housing support expenditure based on monetary subsidies plays a role in promoting the consumption of essential, housing-related and durable goods. This suggests that housing support expenditure based on monetary subsidies is better able to protect the maintenance of necessities.

The contribution of this paper is mainly in the following aspects: first, the theoretical research method expands the research scope of the impact of housing support expenditure on residents' consumption. The existing literature on government housing support and residents' consumption is mostly centered on analyzing the total amount of residents' consumption

expenditure, with less exploring the impact on the structure of residents' consumption. Second, this paper argues that the increase in government housing support expenditure is conducive to promoting the total consumption of urban residents, but the kind of impact on the consumption structure is related to the form of expenditure. The form of expenditure in the form of money is more conducive to boosting residents' consumption of non-essential and non-durable goods, which is more conducive to optimizing the consumption structure. Thirdly, in the econometric study, this paper, in an innovative way, empirically analyzes the consumption effect of two forms of housing support by using the data of guaranteed housing construction and financial housing support expenditure.

## 2. Review of the literature, theoretical investigation, and formulation of hypotheses

### 2.1 Literature review

The literature on the three areas of housing, housing support, and residential consumption is very fruitful. This part of the review focuses on the research results of the three intrinsic linkages while learning from the study to explain the entry point of this paper, the internal logic, and the expected goals of the study.

   **(1) Housing and household consumption.**   The relationship between housing and household consumption is an important topic worldwide. Housing is a long-term asset. A change in the price of this asset for households owning housing will change the nominal or real income of such households [4] and ultimately cause changes in their consumption [5]. Most scholars believe that commercial housing affects consumption through a wealth effect, and one of the transmission paths of this effect is house prices [6]. Changes in house prices may trigger rational optimism and pessimism in the economy [7], which are transmitted to residents' consumption behavior. In the early period, the wealth effect of housing was not significant, and consumption was not highly sensitive to house prices [8]. In recent years, with the real estate market boom, changes in housing prices significantly affect total household consumption expenditures. Using a large amount of Chinese household survey data, scholars have investigated the impact of housing appreciation on urban household consumption. The results show that every 10% increase in housing wealth increases household consumption by about 3% [9]. Other scholars also discovered that the appreciation of housing in three countries, namely, Australia, Canada, and New Zealand, further smoothes consumption, and among these three countries, Canadian residents' consumption is more sensitive to housing appreciation [10]. However, a smaller number of scholars have argued that when factors such as household indebtedness are added, the impact of housing prices on household consumption expenditures can have different results: some scholars have argued that, due to the existence of the investment attributes of housing, not only do rising housing prices not bring about a wealth effect, but they may also have a crowding-out effect on consumption. Using data on China's credit card and debit card transactions from 2011 to 2013, we measured the impact of housing price changes on urban household consumption and found that for every 10 percent increase in housing prices, non-housing consumption would decrease by 9 percent [11]. The effect of housing wealth on household consumption is not particularly significant compared to financial wealth and income mobility [12], and it is also possible that housing prices have different outcomes for household consumption because of different levels of socio-economic development between regions [13]. As the price of commercial housing in China rose, housing rents also rose rapidly, with housing rents having a more significant wealth effect on homeowners [14]. Changes in housing prices also affect the structure of household consumption expenditures. The ups and downs in housing prices explain half of the corresponding fluctuations in

non-durable expenditures through wealth wills [15], and there is an intertemporal dependence between total household non-durable consumption goods and housing [16]. Undeniably, when house prices rise, the cost of housing increases for low- and middle-income earners who rent or are about to buy a home, affecting their household consumption expenditure [17].

  **(2) Housing support and consumption.**   The theme of this paper is government housing support and residents' consumption. Providing housing support for low- and middle-income families can expand urban residents' housing and consumption demands, which has a greater impact on families and the country [18]. There are basically two models of housing support in various countries: one is housing in kind [19], and the other is housing subsidy [20, 21]. Regardless of the type of housing support policy, it aims to improve housing affordability and guarantee residents' quality of life [22]. In terms of applying for subsidized housing, the government will set up some thresholds and conditions to protect the housing rights of low- and middle-income people [23]. Public housing, as the main basis of housing support policy, is an important part of national welfare [24]; as such, South Korea's government aims to improve the housing welfare of low- and middle-income households by implementing public rental housing programs and cash subsidies [25], which include one-person households, newlywed households, etc., and, through public housing provision, the burden of housing costs for all types of households is alleviated, household residence satisfaction is enhanced, and household consumption is promoted [26]. For some specific groups, life support is also provided through public housing [27]. Government-provided public housing will obviously lack in hardware conditions compared to market-distributed commercial housing, which will give rise to the possibility that residents may incur higher consumption of durable goods, such as electrical appliances, and electricity consumption [3], and this group of people has a higher intensity of energy use and consumption [28–30]. Another housing support measure that has a significant impact on residential consumption, monetary subsidies for housing, has a broad impact on residential consumption, with the public housing program and the Housing Choice Voucher Program (HCVP) in the United States being associated with dietary consumption by users [2]. Rental subsidies in Finland and the UK impact rent and house prices [31], and countries with emerging economies establish long-term collective savings schemes to support housing consumption, such as China's Housing Provident Fund (HPF) system [13]. China's HPF system is a system arising from the commoditization of housing to explain the housing problems of urban households with a tripartite commitment theme of government, units, and individuals who conspire to work together to explain the housing problems of urban households, and it is the largest public housing program in China [32]. China's housing provident fund is an important form of housing support, and financial support of a housing provident fund plays a positive role in residents' consumption in general. Also, financial support can significantly reduce the pressure on household housing [13]. Some scholars have also pointed out that it is debatable whether housing subsidies can be considered as a form of income and their impact on recipients' consumption demand [33].

## 2.2 Theoretical analysis

Based on the above literature analysis, this paper argues that government housing support can be transmitted to household consumption through two paths. One is the income effect of housing monetary subsidies, and the other is the employment promotion and commodity housing price calming effect of housing construction. Housing monetary subsidies, as exemplified by China's housing provident fund system, increase consumers' purchasing power on the demand side, characteristics similar to those of the housing subsidy program. The housing provident fund can be considered a form of income for households, and an increase in housing

subsidies by the government can theoretically be considered an increase in residents' direct or indirect income, thus promoting household consumption [34]. As a housing social support system, expanding the coverage of the housing fund is conducive to mobilizing housing funds and increasing the housing affordability of the beneficiary population [35]. The rental subsidy for the renting population can be considered as a direct income increase for the beneficiaries, and there may be a tendency for consumption expenditure to expand after the burden of renting is reduced. The other is the construction of guaranteed housing. The path of the impact of the construction of guaranteed housing on residents' consumption is more complicated. Guaranteed housing is a special nature of housing provided for specific groups of people, with the government as the main investment and construction. Guaranteed housing construction requires a large number of employed people to engage in related projects, which, to a large extent, can lead to local employment. At the same time, Guaranteed housing construction and renovation require large volumes of materials, which can generally lead to the development of the manufacturing and service industries. Also, employing local people in the related industries will produce a driving effect, promote the increase in income of related industries, and ultimately lead to the demand for consumption of residents. The government, as the main body of the construction of support housing, complements commodity housing. Large-scale support housing supply helps to play a positive role in the wealth effect of commodity housing prices and thus improves residents' consumption. Based on the above literary and theoretical analysis, this paper finds that scholars have discussed the consumption role of housing support inadequately and imperfectly. What is the impact of housing support on residents' consumption? What are the specific paths? Do different forms of housing support have different impacts on residents' consumption? What is the impact of housing support on the consumption structure of the population? The analysis of this paper is based on the above questions, based on the data on China's guaranteed housing from 1999 to 2009 and the data on financial expenditures on housing support from 2000 to 2021. This paper uses the system GMM method to analyze the impact of housing support on urban residents' consumption, analyze residents' total consumption and consumption structure, and compare the two forms of support—to dig deeper into the issues related to housing support from the theoretical and empirical perspectives and make up for the lack of research in the current literature.

Based on the above analysis, the hypotheses of this paper are proposed:

Hypothesis 1: Other things being equal, government housing support expenditure impacts urban residents' consumption expenditure.

Hypothesis 2: Different forms of housing support expenditures have different results on urban residents' consumption expenditures.

## 3. Research design

### 3.1 Sample selection and data sources

In order to confirm the effect of government housing support expenditures on the consumption structure of urban residents, this study chooses two-stage panel data for analysis. The Chinese government's housing support funding changed after 2010. Therefore, this development was taken into consideration while choosing the statistics. 1999–2009 housing support expenditure data are mainly selected from the amount of the Chinese government's investment in the construction of affordable housing during this period, and 2010–2021 housing support expenditure data are mainly selected from the data of China's financial expenditure on housing support. China's financial spending on housing assistance is the primary source

of the data on housing support expenditures for 2010–2021. In view of data integrity considerations, the Tibet Autonomous Region is removed from the sample space in this paper. The study used a cross-sectional sample, and since the Shanghai Municipality abolished the affordable housing development center in 2002 and continued to safeguard housing until 2009, the data in the study is incomplete and discontinuous. Lastly, with the exception of Shanghai, Tibet, Hong Kong, Macao, and Taiwan, this study chooses 29 Chinese provinces—that is, municipalities directly under the control of the central government and autonomous regions—as the sample space. All of the data are taken from the National Bureau of Statistics of China website for the relevant years, the China Statistical Yearbook, the China Financial Yearbook, the China Population and Employment Statistical Yearbook, the Wind Information Database, the EPS Global Statistics Database, etc., using 1998 as the base period to lessen the impact of price factors.

## 3.2 Variable setting

**3.2.1 Explained variables.**   Consumption by urban residents ($Consume^{0,1,2,3,4,5,6}$). This article primarily examines the economic impact of housing assistance expenditures from the standpoint of consumption structure, providing an explanation based on the amounts of total consumption expenditures made by urban inhabitants and the structure of those expenditures. The explained variables are selected as total urban per capita consumption ($Consume^0$) and urban per capita consumption structure ($Consume^{1,2,3,4,5,6}$). Based on previous articles, this study divides the consumption structure of urban residents into housing-related goods consumption expenditure ($Consume^1$) and non-housing related goods consumption expenditure ($Consume^2$), essential goods consumption expenditure ($Consume^3$) and non-essential goods consumption expenditure ($Consume^4$), durable goods consumption expenditure ($Consume^5$) and non-durable goods consumption expenditure ($Consume^6$).

**3.2.2 Core explanatory variables.**   Housing support expenditure (Hse). The core explanatory variable of this paper is government housing support expenditure (Hse). The data used in this study pertains to affordable housing and is split into two periods: the housing support expenditure data from 1999 to 2009 and the government housing support financial expenditure data from 2010 to 2021 at the provincial level in China.

**3.2.3 Control variables.**   The income of inhabitants (Income). The theory of consumption contends that residents' consumption is mostly influenced by their income. Therefore, in this study, the indicator of residents' income is the urban per capita disposable income.

The dependence ratio of urban households (Bring). This study chooses the ratio of the total of the minor population (under 14 years old, including 14 years old) and the elderly population (above 65 years old, including 65 years old) to the working-age population (15–64 years old) [36].

The percentage of secondary industry (Industry) and the percentage of tertiary industry (Service). The proportion of secondary industry (Industry) is the ratio of secondary industry GDP to regional GDP [37]; the ratio of the tertiary sector's GDP to the regional GDP is known as the tertiary industry proportion (service) [38].

Average price of commercial residential housing (Chp). With the fluctuation in housing prices in recent years, the cost of commercial housing has also grown to be a significant factor influencing urban residents' consumption [39–41]. Therefore, in this paper, the average cost of commercial real estate in each province and city (Chp-commercial housing pricing) is chosen as a measure of house price.

The detailed description of each variable is shown in Table 1.

**Table 1. Detailed description of each variable.**

| Variable type | variable symbol | variable name | unit of variability | Variable Meaning |
|---|---|---|---|---|
| explanatory variable | lnConsume[0] | total per capita consumption expenditure of urban households | CNY/person | Total urban per capita consumption expenditure in logarithmic terms |
| | lnConsume[1] | urban per capita consumption expenditure on housing-related goods in real terms | CNY/person | Logarithm of the sum of urban per capita consumption expenditures on housing and household equipment and services |
| | lnConsume[2] | urban per capita consumption expenditure on non-housing related goods in real terms | CNY/person | Logarithm of the sum of urban per capita expenditures on food, clothing, transportation and communication, health care, culture, education and recreation, and other expenditures |
| | lnConsume[3] | urban per capita consumption expenditure on essential goods in real terms | CNY/person | Logarithmic value of the sum of urban per capita expenditures on housing, food, clothing, transportation, and communication |
| | lnConsume[4] | urban per capita consumption expenditure on non-essential goods in real terms | CNY/person | Logarithm of the sum of urban per capita expenditures on health care, household equipment and services, culture, education and recreation, and other expenditures |
| | lnConsume[5] | urban per capita consumption expenditure on durable goods in real terms | CNY/person | Logarithm of the sum of urban per capita household consumption expenditures on consumer durables, interior decorations, furniture materials, bedding, household miscellaneous goods, medical and healthcare appliances, household transportation, communication tools, cultural and recreational goods, and housing |
| | lnConsume[6] | urban per capita consumption expenditure on non-durable goods in real terms | CNY/person | Total urban per capita consumption minus consumption expenditure on durable goods, logarithmic value |
| Main explanatory variables | lnHse | Real value of per capita investment in affordable housing in towns and cities (1999–2009) Urban per capita housing support expenditure in real terms (2010–2021) | CNY/person | Urban per capita investment in affordable housing in logarithmic terms (1999–2009) Logarithmic urban per capita financial expenditure on housing support (2010–2021) |
| control variable | lnIncome | Real disposable income per urban resident | CNY/person | Urban disposable income per capita in logarithmic terms |
| | lnChp | Average residential sales price of commercial properties in real terms | CNY/square meter | Logarithmic average sales price of commercial residential units |
| | Service | Tertiary sector to GDP ratio | % | Value added of tertiary sector/regional GDP |
| | Industry | Secondary sector to GDP ratio | % | Value added of secondary industry/regional GDP |
| | Bring | Urban household dependency ratio | % | (Population up to and including 14 years of age + population over 65 years of age including 65 years of age)/Population 15–64 years of age |

## 3.3 Modeling

The following econometric model form is set up in conjunction with this research's content and objectives:

$$\ln Consume_{it}^{(k)} = \beta_0^{(k)} + +\beta_1^{(k)} \ln Hse_{it} + \gamma^{(k)} X_{it} + \alpha_i^{(k)} + \alpha_t^{(k)} + \varepsilon_{it}^{(k)}$$

Where the subscripts i and t denote the ith province (city) and the tth year, respectively, the six values of k = 0,1,2,3,4,5,6 represent the total per capita housing-related goods consumption expenditure, per capita non-housing related goods consumption expenditure, per capita essential goods consumption expenditure, per capita non-essential goods consumption expenditure, per capita consumption expenditure on durable goods, per capita consumption expenditure on non-durable goods, respectively. The above variables all appear in the model in logarithmic form. $\ln Hse_{it}$ is the main observation variable, which utilizes the per capita affordable housing investment data in each province from 1999 to 2009 and the financial expenditure data on housing support in each province from 2010 to 2021. The data for both time periods are taken

**Table 2. Statistical description of the main variables (1999–2009).**

| variable symbol | Variable Meaning | average value | standard deviation | minimum value | maximum values | inter-quartile rangee | Number of observations |
|---|---|---|---|---|---|---|---|
| lnConsume[0] | total per capita consumption expenditure of urban households | 8.780271 | 0.3177255 | 8.148623 | 9.646826 | 0.435967 | 319 |
| lnConsume[1] | urban per capita consumption expenditure on housing-related goods in real terms | 6.991873 | 0.3164504 | 6.228776 | 7.839439 | 0.470927 | 319 |
| lnConsume[2] | urban per capita consumption expenditure on non-housing related goods in real terms | 8.595432 | 0.3241054 | 7.924436 | 9.495299 | 0.442622 | 319 |
| lnConsume[3] | urban per capita consumption expenditure on essential goods in real terms | 8.362784 | 0.2989137 | 7.741271 | 9.105307 | 0.428906 | 319 |
| lnConsume[4] | urban per capita consumption expenditure on non-essential goods in real terms | 7.699283 | 0.36513 | 6.90383 | 8.774877 | 0.447328 | 319 |
| lnConsume[5] | urban per capita consumption expenditure on durable goods in real terms | 7.586582 | 0.39363 | 6.705511 | 8.660008 | 0.4973194 | 319 |
| lnConsume[6] | urban per capita consumption expenditure on non-durable goods in real terms | 8.415678 | 0.2905836 | 7.851653 | 9.18428 | 0.4168317 | 319 |
| lnHse | urban per capita affordable housing investment in real terms | 4.698569 | 0.8399499 | 1.595801 | 6.923169 | 0.839255 | 319 |
| lnIncome | real disposable income per urban resident | 9.054412 | 0.3485144 | 8.372239 | 10.0485 | 0.494445 | 319 |
| lnChp | average residential sales price of commercial properties in real terms | 7.618805 | 0.4365458 | 6.709304 | 9.393124 | 0.498547 | 319 |
| Service | value added of tertiary sector as a share of GDP | 38.49921 | 6.9665 | 28.6 | 75.5 | 0.0502632 | 319 |
| Industry | value added of the secondary sector as a share of GDP | 46.36041 | 7.0656 | 19.76 | 61.5 | 0.0916 | 319 |
| Bring | urban household dependency ratio | 34.35917 | 4.3563 | 24.15781 | 47.92317 | 0.0573704 | 319 |

in logarithmic form. $X_{it}$ = (ln $Income_{it}$, ln $chp_{it}$, $Service_{it}$, $Industry_{it}$, $Bring_{it}$) as a vector of other variables. ln $Income_{it}$ represents the logarithm of real disposable income per capita of urban residents. ln $chp_{it}$ stands for the Average Sales Price of Commercial Properties Taken in Logarithms. $Industry_{it}$ and $Service_{it}$ represent the ratio of the secondary sector to GDP and the ratio of the tertiary sector to GDP in each province, respectively. $Bring_{it}$ stands for Urban Household Dependency Ratio—the formula is (number of people under 14 and 14 years old + number of people over 65 and 65 years old/number of people aged 15–64). $\alpha_i$ stands for individual fixed effects, $\alpha_i$ stands for year fixed effects, and $\varepsilon_{it}$ stands for the randomized disturbance term. Descriptive statistics for each variable are shown in Tables 2 and 3 below:

Data sources for the pertinent years include the China Financial Yearbook, China Statistical Yearbook, China Population and Employment Statistical Yearbook, Wind Information database, EPS data platform, and data from the Sixth Population Census.

# 4. Empirical findinds

## 4.1 Benchmark regression analysis

For static panel data, there are usually three estimation methods: mixed model (P-OLS), random effects model, and fixed effects model. In order to determine the static benchmark model regression method in this section, this paper carries out the following selection. Firstly, this paper carries out the F-test and finds that the value of the F-statistic is less than 0.05, then it can be known that the fixed effect model is better than the mixed effect model; secondly, this paper applies Hausman's test to select between the fixed effect model and the random effect model, and the results show that the fixed effect is the best choice in this paper. Therefore, this paper chooses the fixed effect model as the benchmark model estimation in this section.

**Table 3. Statistical description of the main variables (2010–2021).**

| variable symbol | Variable Meaning | average value | standard deviation | minimum value | maximum values | inter-quartile range | Number of observations |
|---|---|---|---|---|---|---|---|
| $lnConsume^0$ | total per capita consumption expenditure of urban households | 9.778141 | 0.4160734 | 8.848667 | 10.75312 | 0.6992756 | 348 |
| $lnConsume^1$ | urban per capita consumption expenditure on housing related goods in real terms | 8.317499 | 0.6563921 | 6.997444 | 9.958283 | 1.174945 | 348 |
| $lnConsume^2$ | urban per capita consumption expenditure on non-housing related goods in real terms | 9.499046 | 0.3533877 | 8.670676 | 10.25657 | 0.5970303 | 348 |
| $lnConsume^3$ | urban per capita consumption expenditure on essential goods in real terms | 9.40312 | 0.460322 | 8.474456 | 10.45435 | 0.7843792 | 348 |
| $lnConsume^4$ | urban per capita consumption expenditure on non-essential goods in real terms | 8.603371 | 0.3450136 | 7.684457 | 9.437651 | 0.5469288 | 348 |
| $lnConsume^5$ | urban per capita consumption expenditure on durable goods in real terms | 8.730045 | 0.4622126 | 7.650979 | 10.03187 | 0.6451928 | 348 |
| $lnConsume^6$ | urban per capita consumption expenditure on non-durable goods in real terms | 9.343125 | 0.4007842 | 8.489286 | 10.11786 | 0.7145432 | 348 |
| lnHse | urban per capita housing support expenditure in real terms | 6.371385 | 0.6612878 | 3.944434 | 7.924205 | 0.7226894 | 348 |
| lnIncome | real disposable income per urban resident | 10.16609 | 0.4429112 | 9.214114 | 11.30857 | 0.7599736 | 348 |
| lnChp | average residential sales price of commercial properties in real terms | 8.657214 | 0.5451668 | 7.685746 | 10.75665 | 0.712716 | 348 |
| Service | tertiary value added/GDP | 46.18817 | 9.52327 | 5.184927 | 83.86823 | 11.5105 | 348 |
| Industry | value added of secondary sector/GDP | 43.54082 | 8.745879 | 15.8337 | 59 | 10.5211 | 348 |
| Bring | urban household dependency ratio | 33.06135 | 5.557979 | 20.1346 | 46.35434 | 6.998543 | 348 |

This research accounts for individual and year-fixed factors to remove unobserved effects, and Tables 4–6 display the benchmarks' regression findings. The total per-capita consumption costs of urban households (lnConsume0) is the independent variable in Table 4 below. The regression results show that increasing government spending on housing support in any way greatly increases the overall consumption spending of urban residents. The model Fe1-Fe4 in the Table 4 shows the fixed effects regression results. Models Fe1 and Fe2 show that for every 1% increase in funding for affordable housing, urban households' total per capita consumption expenditure rises by 0.019%–0.045%. Models Fe3 and Fe4 show that for every 1% increase in government financial housing support expenditure, urban residents' total per capita consumption expenditure rises by 0.027%–0.028%.

**Table 4. Regression results of the impact of government housing support expenditure on total per capita consumption expenditure of urban residents (1999–2021).**

| | 1999–2009 | | 2010–2021 | |
|---|---|---|---|---|
| | Fe1 | Fe2 | Fe3 | Fe4 |
| lnHse | 0.045* (0.029) | 0.019* (0.010) | 0.028*** (0.007) | 0.027*** (0.007) |
| lnIncome | | | 0.907*** (0.008) | 0.950*** (0.023) |
| lnChp | | 0.545*** (0.031) | | -0.004 (0.026) |
| Bring | | -0.006** (0.002) | | -0.005*** (0.001) |
| service | | 0.031*** (0.004) | | 0.000 (0.001) |
| industry | | 0.034*** (0.003) | | 0.000 (0.001) |
| constant term | 8.569*** (0.135) | 1.933*** (0.215) | 0.375*** (0.061) | 0.111 (0.167) |
| sample size | 319 | 319 | 348 | 348 |
| Within-$R^2$ | 0.850 | 0.886 | 0.987 | 0.976 |

Note: Hereinafter ***, **, and * represent significance at the 1%, 5%, and 10% levels, respectively. Standard errors are in parentheses.

**Table 5. Regression results of the impact of government housing support expenditure on the consumption structure of urban residents (1999–2009).**

| | 1999–2009 | | | | | |
| | lnConsume[1] (housing) | lnConsume[2] (non-housing) | lnConsume[3] (essential) | lnConsume[4] (non-essential) | lnConsume[5] (durable) | lnConsume[6] (non-durable) |
| | Fe1 | Fe2 | Fe3 | Fe4 | Fe5 | Fe6 |
|---|---|---|---|---|---|---|
| lnHse | 0.257* (0.214) | 0.138** (0.068) | 0.088* (0.069) | 0.299** (0.135) | 0.118* (0.169) | 0.179*** (0.066) |
| lnIncome | 0.844*** (0.063) | 0.861*** (0.020) | 0.899*** (0.020) | 0.781*** (0.040) | 1.011*** (0.049) | 0.794*** (0.019) |
| lnChp | -0.035 (0.113) | 0.071** (0.036) | -0.004 (0.036) | 0.153** (0.071) | -0.058 (0.089) | 0.095*** (0.035) |
| Bring | 1.731** (0.862) | 0.264 (0.273) | 0.604** (0.276) | 0.493 (0.544) | 0.442 (0.681) | 0.573** (0.267) |
| lnHselnChp | -0.020 (0.022) | -0.013* (0.007) | -0.008 (0.007) | -0.026* (0.014) | -0.006 (0.017) | -0.018*** (0.007) |
| lnHseBring | -0.302 (0.192) | -0.107* (0.061) | -0.095 (0.061) | -0.255** (0.121) | -0.210 (0.151) | -0.120** (0.059) |
| Service | 0.128 (0.478) | 0.130 (0.152) | -0.150 (0.153) | 0.688** (0.302) | 1.064*** (0.378) | -0.209 (0.148) |
| Industry | 0.333 (0.434) | 0.173 (0.138) | 0.031 (0.139) | 0.539* (0.274) | 0.796** (0.343) | -0.015 (0.135) |
| constant term | -1.186 (1.020) | 0.022 (0.324) | 0.111 (0.327) | -1.284** (0.643) | -2.052** (0.806) | 0.381 (0.317) |
| observed value | 319 | 319 | 319 | 319 | 319 | 319 |
| Within-R$^2$ | 0.835 | 0.986 | 0.985 | 0.947 | 0.939 | 0.984 |

## 4.2 Heterogeneity analysis

**4.2.1 Baseline regression.**    This paper divides the consumption structure into three perspectives to learn more about how government housing support spending affects urban dwellers' purchasing patterns: (1) household consumption expenditure on housing and non-housing, (2) household consumption expenditure on essential and non-essential, and (3) household consumption expenditure on durable and non-durable goods. The estimation results of the static model are shown in Tables 5 and 6. In Table 5, using the investment amount of affordable housing as the dependent variable, the effect of housing support spending on the variability of the consumption structure of urban residents at the level of "making up for the bricks" is examined. It is clear from the table that the government's building of such dwellings will have a favorable impact on the growth of cheap housing, regardless of how It is divided.

**Table 6. Static regression results of the impact of government housing support expenditure on the consumption structure of urban residents (2010–2021).**

| | 2010–2021 | | | | | |
| | lnConsume[1] (housing) | lnConsume[2] (non-housing) | lnConsume[3] (essential) | lnConsume[4] (non-essential) | lnConsume[5] (durable) | lnConsume[6] (non-durable) |
| | Fe1 | Fe2 | Fe3 | Fe4 | Fe5 | Fe6 |
|---|---|---|---|---|---|---|
| lnHse | 0.164*** (0.027) | -0.018* (0.01) | 0.083*** (0.01) | -0.089*** (0.017) | 0.047*** (0.015) | 0.008 (0.011) |
| lnIncome | 1.552*** (0.083) | 0.816*** (0.032) | 1.098*** (0.031) | 0.702*** (0.035) | 0.984*** (0.031) | 0.936*** (0.023) |
| lnChp | -0.210** (0.091) | 0.000 (0.037) | -0.053 (0.035) | | | |
| Bring | | -0.006*** (0.001) | -0.004*** (0.002) | -0.004* (0.002) | 0.002 (0.002) | -0.009*** (0.001) |
| Service | | -0.000 (0.001) | 0.001 (0.001) | -0.001 (0.002) | 0.001 (0.002) | -0.000 (0.001) |
| Industry | | -0.001 (0.002) | 0.004** (0.002) | -0.007*** (0.003) | 0.006** (0.002) | -0.003 (0.002) |
| constant term | -6.687*** (0.220) | 1.594*** (0.232) | -1.882*** (0.223) | 2.515*** (0.382) | -1.952*** (0.336) | 0.183 (0.253) |
| observed value | 348 | 348 | 348 | 348 | 348 | 348 |
| Within-R$^2$ | 0.939 | 0.967 | 0.984 | 0.892 | 0.956 | 0.971 |

Robust standard errors are in parentheses.

Table 6 shows the results of the static regression with government financial housing support expenditure as the dependent variable, i.e., investigating how housing support spending affects the variation in the consumption patterns of urban people at the "headcount" level. From the models Fe1, Fe3, and Fe5, it is concluded that increasing government financial housing assistance spending can greatly enhance urban inhabitants' spending on housing-related necessities for their homes, thus playing the role of "preserving the foundation and promoting consumption". From Models Fe2, Fe4, and Fe6, the government financial housing support expenditure increase for urban residents to enjoy some of the consumption does not play a significant role. Moreover, non-housing consumption and non-essential consumption have a significant negative effect.

**4.2.2 Analysis based on housing and non-housing consumption expenditures.** The regression analysis results above are obtained using the static model in this paper. Next, to re-estimate the dependent variable, this research builds a dynamic model and integrates the lagged period of the dependent variable into the regression model. At this point, the model has some endogenous problems. Considering the endogeneity problem of the explanatory variables and the constraints of the short panel data, the use of the fixed effects model (FE) cannot overcome the endogeneity problem, thus failing to obtain an effective unbiased estimator, at which point generalized moment estimation becomes the optimal choice. As for the specific application of the generalized moments estimation method, Blundell & Bond (1998) believe that systematic moments estimation (SYS-GMM) under certain conditions is more accurate than differential moments estimation (DIF-GMM). Therefore, this paper uses systematic moment estimation (SYS-GMM) for estimation. To ensure the accuracy and reasonableness of the estimation results, the Hansen test value is used to determine whether there is an over-identification of instrumental variables, and the AR(1) and AR(2) test values are used to determine whether the residual terms are autocorrelated or not [42–44]. Tables 7–9 show the system GMM regression results of housing and non-housing consumption expenditure, essential

**Table 7. Results of GMM regression of housing and non-housing consumption expenditure system of structure.**

|  | 1999–2009 | | | | 2010–2021 | | | |
|---|---|---|---|---|---|---|---|---|
|  | Housing (lnConsume[1]) | | non-housing (lnConsume[2]) | | Housing (lnConsume[1]) | | non-housing (lnConsume[2]) | |
|  | Gmm1 | Gmm2 | Gmm3 | Gmm4 | Gmm5 | Gmm6 | Gmm7 | Gmm8 |
| L.lnConsume[1] | 0.734*** (0.057) | 0.697*** (0.053) |  |  | 0.899*** (0.094) | 0.150** (0.070) |  |  |
| L.lnConsume[2] |  |  | 0.502*** (0.104) | 0.510*** (0.073) |  |  | 0.037 (0.018) | 0.026 (0.020) |
| lnHse | 0.010** (0.007) | 0.608** (0.269) | 0.006 (0.006) | 0.137 (0.094) | 0.124* (0.071) | 0.153** (0.062) | 0.111 (0.015) | -0.013** (0.015) |
| lnIncome | 0.217*** (0.040) | 0.221*** (0.047) | 0.458*** (0.094) | 0.429*** (0.067) |  | 2.159*** (0.152) | 0.788*** (0.041) | 0.890*** (0.026) |
| lnChp |  | 0.282*** (0.102) |  | 0.091* (0.048) | -0.073 (0.137) | -0.770*** (0.122) | -0.005 (0.045) |  |
| Bring |  | 2.325 (1.571) |  | -0.159 (0.415) |  | -0.008** (0.004) |  | -0.007*** (0.002) |
| lnHseBring |  | -0.470 (0.338) |  | 0.007 (0.088) |  |  |  |  |
| lnHselnChp |  | -0.057** (0.023) |  | -0.017* (0.009) |  |  |  |  |
| Service |  | 0.121 (0.138) |  | 0.118 (0.103) |  | 0.002 (0.003) |  | 0.001 (0.001) |
| Industry |  | 0.193* (0.112) |  | 0.032* (0.063) |  | -0.018** (0.006) |  | -0.001 (0.002) |
| constant term | -0.102 (0.129) | -2.990** (1.245) | 0.143 (0.092) | -0.338 (0.458) | 0.145* (0.236) | 1.785 (1.647) | 0.634* (1.753) | 0.452 (0.863) |
| observed value | 290 | 290 | 290 | 290 | 290 | 290 | 290 | 290 |
| AR (1)-p | 0.000 | 0.000 | 0.000 | 0.000 | 0.003 | 0.000 | 0.000 | 0.000 |
| AR (2)-p | 0.333 | 0.525 | 0.415 | 0.496 | 0.446 | 0.339 | 0.012 | 0.084 |
| Hansen-p | 0.339 | 0.299 | 0.398 | 0.566 | 0.998 | 0.988 | 0.995 | 0.993 |

Robustness standard errors are in parentheses, and p-values are shown for each test

**Table 8. Systematic GMM regression results of structural essential and non-essential consumption expenditures.**

| | 1999–2009 | | | | 2010–2021 | | | |
|---|---|---|---|---|---|---|---|---|
| | lnConsume[3] (essential) | | lnConsume[4] (non-essential) | | lnConsume[3] (essential) | | lnConsuem[4] (non-essential) | |
| | Gmm1 | Gmm2 | Gmm3 | Gmm4 | Gmm5 | Gmm6 | Gmm7 | Gmm8 |
| L.lnConsume[3] | 0.568*** (0.071) | 0.604*** (0.094) | | | 0.313*** (0.045) | 0.050*** (0.055) | | |
| L.lnConsume[4] | | | 0.850*** (0.058) | 0.706*** (0.058) | | | 0.062 (0.066) | 0.125 (0.074) |
| lnHse | 0.007* (0.003) | 0.150** (0.064) | 0.005 (0.006) | 0.153 (0.133) | 0.228*** (0.057) | 0.073*** (0.019) | -0.071** (0.028) | -0.086** (0.029) |
| lnIncome | 0.371*** (0.06) | 0.356*** (0.082) | 0.144** (0.054) | 0.209*** (0.054) | | 1.284*** (0.018) | 0.560*** (0.072) | 0.570*** (0.065 |
| lnChp | | 0.065* (0.031) | | 0.116* (0.062) | 0.600*** (0.049) | -0.193*** (0.05) | 0.12 (0.079) | 0.198** (0.069) |
| Bring | | 0.38 (0.256) | | 0.305 (0.629) | | -0.007*** (0.001) | | -0.007** (0.002) |
| lnHse*Bring | | -0.083 (0.059) | | -0.091 (0.139) | | | | |
| lnHse*lnChp | | -0.014** (0.006) | | -0.016 (0.013) | | | | |
| Service | | -0.131* (0.071) | | 0.308** (0.132) | | -0.001* (0.000) | | 0.001 (0.003) |
| Industry | | -0.062 (0.053) | | 0.124 (0.093) | | -0.002 (0.002) | | 0.006* (0.003) |
| constant term | 0.269** (0.114) | -0.479* (0.26) | -0.107 (0.095) | -0.729 (0.592) | 0.324** (0.467) | 0.927** (0.126) | 0.843** (0.642) | 0.395 (0.435) |
| observed value | 290 | 290 | 290 | 290 | 290 | 290 | 290 | 290 |
| AR (1)-p | 0.000 | 0.000 | 0.000 | 0.000 | 0.000 | 0.008 | 0.000 | 0.001 |
| AR (2)-p | 0.333 | 0.525 | 0.415 | 0.496 | 0.038 | 0.113 | 0.217 | 0.186 |
| Hansen-p | 0.339 | 0.299 | 0.398 | 0.566 | 0.997 | 0.995 | 0.995 | 0.993 |

Robustness standard errors in parentheses; p-values for each test are reported below

and non-essential consumption expenditure, and durable and non-durable consumption expenditure, respectively.

In Tables 7–9, the level of government investment in affordable housing, or the dependent variable, is what separates the 1999–2009 data set. Housing support, also known as "making up for bricks and mortar", and the dependent variable for the 2010–2021 data is the government's

**Table 9. Systematic GMM regression results of consumption expenditures on durable and non-durable goods by structure.**

| | 1999–2009 | | | | 2010–2021 | | | |
|---|---|---|---|---|---|---|---|---|
| | lnConsume[5] (durable) | | lnConsume[6] (non-durable) | | lnConsume[5] (durable) | | lnConsume[6] (non-durable) | |
| | Gmm1 | Gmm2 | Gmm3 | Gmm4 | Gmm5 | Gmm6 | Gmm7 | Gmm8 |
| L.lnConsume[5] | 0.557*** (0.078) | 0.436*** (0.067) | | | 0.965*** (0.048) | 0.629*** (0.078) | | |
| L.lnConsume[6] | | | 0.820*** (0.073) | 0.834*** (0.064) | | | 0.014 (0.026) | 0.015 (0.024) |
| lnHse | 0.013* (0.009) | 0.317** (0.140) | 0.005 (0.002) | 0.046 (0.048) | 0.054** (0.025) | 0.015* (0.024) | 0.022 (0.014) | 0.017 (0.016) |
| lnIncome | 0.449*** (0.074) | 0.491*** (0.080) | 0.165*** (0.059) | 0.167*** (0.057) | | 0.582*** (0.082) | 1.040*** (0.053) | 1.033*** (0.031) |
| lnChp | | 0.178** (0.075) | | 0.022 (0.022) | -0.081* (0.053) | -0.286*** (0.055) | -0.065 (0.075) | |
| Bring | | 1.115 (0.713) | | -0.179 (0.238) | | | -0.011*** (0.001) | -0.011*** (0.002) |
| lnHse*Bring | | -0.225 (0.167) | | 0.013 (0.050) | | | | |
| lnHse*lnChp | | -0.030** (0.015) | | -0.006 (0.005) | | | | |
| Service | | 0.532*** (0.132) | | -0.115** (0.053) | | | | 0.000 (0.001) |
| Industry | | 0.285** (0.107) | | -0.056* (0.033) | | 0.002 (0.002) | | 0.003* (0.002) |
| constant term | -0.708** (0.149) | -2.245** (0.638) | 0.047 (0.096) | -0.128 (0.248) | -0.541** (0.368) | 0.735* (0.325) | 0.675** (0.162) | 0.463 (0.361) |
| observed value | 290 | 290 | 290 | 290 | 290 | 290 | 290 | 290 |
| AR(1)-p | 0.000 | 0.000 | 0.000 | 0.000 | 0.000 | 0.000 | 0.000 | 0.002 |
| AR(2)-p | 0.510 | 0.609 | 0.353 | 0.498 | 0.998 | 0.598 | 0.334 | 0.009 |
| Hansen-p | 0.598 | 0.443 | 0.625 | 0.441 | 0.996 | 0.994 | 0.997 | 0.992 |

Robustness standard errors in parentheses; p-values for each test are reported.

fiscal expenditure on housing support, i.e., housing support in the form of "making up for heads and mortar".

The analysis of Table 7 is as follows: Models Gmm1 and Gmm2 and Models Gmm5 and Gmm6 show that, regardless of the form of housing support expenditure, there is a significant positive relationship with urban residents' housing consumption expenditure. Models Gmm1 and Gmm2 show that every 1% increase in government investment in affordable housing can effectively increase urban residents' housing consumption expenditures by 0.01%-0.608%. The models Gmm5 and Gmm6 demonstrate that a 1% increase in government financial housing support spending can effectively encourage a 0.124%–0.153% increase in the amount that urban households spend on housing. Furthermore, there is no discernible or even a negative impact of government housing support spending on non-housing consumption. Specifically, models Gmm3 and Gmm4 show that increasing government investment in affordable housing cannot significantly promote urban residents' non-housing consumption. Models Gmm7 and Gmm8 demonstrate that, to some extent, an increase in government financial housing support expenditure will negatively impact urban residents' non-housing consumption expenditure, i.e., it will cause a decrease in non-housing consumption expenditure.

**4.2.3 Analysis based on essential and non-essential consumption expenditures.**    Table 8 yields the following analysis: Models Gmm1, Gmm2, and Models Gmm5 and Gmm6 yield that no matter what form of housing support expenditure form, it is significantly positively related to urban residents' necessity expenditure. Models Gmm1 and Gmm2 show that every 1% increase in government investment in affordable housing can effectively increase urban residents' necessity consumption expenditures by 0.007%-0.15%. Models Gmm5 and Gmm6 show that every 1% increase in the government's financial housing support expenditures can effectively increase urban residents' housing consumption expenditures by 0.073%-0.228%. Different from the above results, it is derived from Models Gmm3, Gmm4, and Models Gmm7, Gmm8 that the two forms of housing support expenditures have a non-significant or even negative impact on urban residents' non-essential consumption expenditures. Models Gmm3 and Gmm4 specifically show that an increase in government investment in affordable housing has a favorable but negligible impact on the rise in consumption expenditures of urban inhabitants, i.e., it doesn't really make a difference. From Models Gmm7 and Gmm8, the government's increase in housing support expenditures has a significantly detrimental influence on urban inhabitants' non-essential consumption expenditures. That is, the government expansion of the financial housing support expenditure increase can not effectively promote the urban residents' "enjoyment" of life. This also reflects that, compared with the "brick and mortar" approach, the "headcount" approach to housing support spending can better fulfill the policy's role of preserving the basics and ensuring the necessities.

**4.2.4 Analysis based on consumption of durable goods versus consumption expenditure on nondurable goods.**    Table 9 yields the following analysis: from Models Gmm1, Gmm2 and Models Gmm5, Gmm6, it follows that regardless of the form of housing support expenditure form, there is a significant positive relationship with urban residents' durable goods consumption expenditure. According to models Gmm1 and Gmm2, a 1% increase in government funding for affordable housing can effectively encourage a 0.013%–0.317% increase in urban residents' spending on durable goods. Models Gmm5 and Gmm6 show that every 1% increase in the government's financial housing support expenditures can effectively increase urban residents' durable goods consumption expenditures by 0.015%-0.054%. Unlike the above results, the two forms of housing support expenditures, as derived from models Gmm3 and Gmm4 and models Gmm7 and Gmm8, don't significantly affect the spending on non-essential purchases by city dwellers.

## 4.3 Model robustness tests

Finding out if adjustments to the parameter settings have an impact on the model's output is the goal of the model robustness test [45, 46]. When modifications to the parameter values have no discernible impact on the sign or significance of the variables, the set model is considered robust. When the model is not robust, there needs to be some readjustments. There are three main methods for model robustness testing: multiple regression control method, variable substitution method and adjusted data method. The robustness test that is tested by using multiple measures regression has been demonstrated in this paper. For example, in the previous content of this chapter, when modifications to the parameter choices have little effect on the sign or significance of the variables, the set model is considered resilient. The set model finds that the sign and significance of the core variables have not changed much, while other variables have only slight variations. It is resilient when changes to the parameter settings do not significantly influence the sign or significance of the variables. The econometric approach demonstrates the robustness of this paper's estimation findings.

This study alters the data by removing the three municipalities of Beijing, Tianjin, and Chongqing to repeat the regression. In the tests that were already conducted, this paper removed Shanghai, so the remaining three municipality samples are removed here, with the outcome shown in Tables 10–13.

The regression findings using total urban consumption expenditure as the dependent variable are displayed in Table 10. From the table, it can be concluded that the housing support expenditure, whether in the form of "supplementing headcount" or "supplementing bricks", can successfully encourage urban dwellers' overall consumption spending. The outcome is in line with what was discovered in the earlier investigation.

After removing the municipalities of Beijing, Tianjin, and Chongqing, Table 11 displays the test findings for the dependent variables housing and non-housing consumption expenditures. The table indicates that both types of housing support expenditure have a large positive impact on housing consumption expenditure, but neither type significantly affects non-housing consumption. This aligns with the results of the earlier investigation.

Table 12 shows the test results where the dependent variables are essential and non-essential consumption expenditures after excluding the municipalities of Beijing, Tianjin, and Chongqing. The table shows that for essential consumption expenditures, while neither form of housing support expenditures has a substantial impact on them for non-essential

**Table 10. Robustness test results of total urban residents' consumption expenditure.**

|  | 1999–2009 (lnConsume⁰) | 2010–2021 (lnConsume⁰) |
|---|---|---|
| L.lnConsume⁰ | 0.832*** (0.054) | 0.211*** (0.037) |
| lnHse | 0.015** (0.006) | 0.215*** (0.028) |
| lnIncome |  |  |
| lnChp | 0.119** (0.039) | 0.603*** (0.058) |
| Bring | 0.002 (0.001) | -0.009*** (0.002) |
| lnHseBring |  |  |
| lnHselnChp |  |  |
| Service | 0.007* (0.004) | 0.003 (0.003) |
| Industry | 0.006* (0.003) | -0.008 (0.005) |
| observed value | 260 | 260 |
| ar1p | 0.000 | 0.000 |
| ar2p | 0.128 | 0.505 |
| hansenp | 0.421 | 0.598 |

**Table 11. Results of robustness tests for housing and non-housing consumption expenditures of structure.**

| | 1999–2009 2010–2021 | | | |
| | Housing (lnConsume[1]) | non-housing (lnConsume[2]) | Housing (lnConsume[1]) | non-housing (lnConsume[2]) |
| | Gmm1 | Gmm2 | Gmm3 | Gmm4 |
|---|---|---|---|---|
| L.lnConsume[1] | 0.711*** (0.049) | | -0.049 (0.081) | |
| L.lnConsume[2] | | 0.523*** (0.066) | | -0.001 (0.020) |
| lnHse | 0.547* (0.313) | 0.134 (0.092) | 0.196** (0.070) | -0.017 (0.021) |
| lnIncome | 0.209*** (0.048) | 0.399*** (0.059) | 1.862*** (0.167) | 0.818*** (0.051) |
| lnChp | 0.230* (0.117) | 0.105** (0.047) | -0.672*** (0.119) | 0.081 (0.051) |
| Bring | 2.609 (1.853) | -0.199 (0.353) | -0.005 (0.003) | -0.007*** (0.002) |
| lnHseBring | -0.541 (0.403) | 0.005 (0.081) | | |
| lnHselnChp | -0.046 (0.027) | -0.017* (0.010) | | |
| Service | 0.104 (0.191) | 0.233* (0.128) | | |
| Industry | 0.166 (0.108) | -0.013 (0.069) | -0.017** (0.005) | 0.004** (0.002) |
| observed value | 260 | 260 | 260 | 260 |
| ar1p | 0.001 | 0.000 | 0.000 | 0.000 |
| ar2p | 0.234 | 0.170 | 0.135 | 0.339 |
| hansenp | 0.550 | 0.304 | 0.950 | 0.924 |

Robust standard errors in parentheses; p-values for each test are reported.

consumption, both have a considerable positive impact on them for housing support. This is in line with the findings of the prior tests.

Table 13 shows the results of the test where the dependent variables are durable and non-durable consumption expenditures after excluding the municipalities of Beijing, Tianjin, and Chongqing. The table shows that both types of housing support expenditures significantly benefit durable consumption expenditures. However, neither type of housing support expenditure

**Table 12. Robustness test results of structural necessity and non-necessity consumption expenditure.**

| | 1999–2009 | | 2010–2021 | |
| | Necessity (lnConsume[3]) | Non-essential (lnConsume[4]) | Necessity (lnConsume[3]) | Non-essential (lnConsume[4]) |
| | Gmm5 | Gmm6 | Gmm7 | Gmm8 |
|---|---|---|---|---|
| L.lnConsume3 | 0.626*** (0.065) | | 0.041** (0.019) | |
| L.lnConsume4 | | 0.667*** (0.053) | | 0.184** (0.085) |
| lnHse | 0.179* (0.105) | 0.194 (0.189) | 0.087*** (0.019) | 0.009 (0.037) |
| lnIncome | 0.322*** (0.067) | 0.237*** (0.053) | 1.212*** (0.053) | |
| lnChp | 0.088* (0.046) | 0.139* (0.079) | -0.144** (0.050) | 0.618*** (0.077) |
| Bring | 0.457 (0.269) | 0.291 (0.852) | -0.006*** (0.001) | -0.008** (0.002) |
| lnHseBring | -0.115 (0.068) | -0.102 (0.197) | | |
| lnHselnChp | -0.017* (0.012) | -0.021* (0.018) | | |
| Service | -0.088 (0.101) | 0.596*** (0.171) | | 0.003 (0.003) |
| Industry | -0.061 (0.057) | 0.178 (0.097) | -0.001 (0.001) | -0.001 (0.005) |
| observed value | 260 | 260 | 260 | 260 |
| AR(1) | 0.000 | 0.000 | 0.001 | 0.000 |
| AR(2) | 0.702 | 0.843 | 0.232 | 0.929 |
| hansenp | 0.221 | 0.263 | 0.685 | 0.998 |

Robust standard errors are in parentheses; p-values for each test are reported.

**Table 13. Results of robustness tests on durable and non-essential durable consumption expenditures of structure.**

|  | 1999–2009 | | 2010–2021 | |
|---|---|---|---|---|
|  | Durable (lnConsume[5]) | non-durable (lnConsume[6]) | Durable (lnConsume[5]) | non-durable (lnConsume[6]) |
|  | Gmm9 | Gmm10 | Gmm11 | Gmm12 |
| L.lnConsume5 | 0.405*** (0.078) |  | 0.967*** (0.054) |  |
| L.lnConsume6 |  | 0.773*** (0.083) |  | 0.009 (0.024) |
| lnHse | 0.308 (0.188) | 0.079 (0.049) | 0.088** (0.029) | 0.018 (0.025) |
| lnIncome | 0.511*** (0.099) | 0.201*** (0.070) |  | 1.036*** (0.035) |
| lnChp | 0.188* (0.095) | 0.050 (0.018) | -0.077* (0.049) |  |
| Bring | 0.939 (0.889) | -0.159 (0.235) | -0.002 (0.002) | -0.011*** (0.002) |
| lnHseBring | -0.205 (0.213) | -0.002 (0.052) |  |  |
| lnHselnChp | -0.030 (0.021) | -0.010* (0.005) |  |  |
| Service | 0.517** (0.217) | -0.041 (0.081) | -0.001(0.001) | -0.000 (0.001) |
| Industry | 0.266** (0.110) | -0.049 (0.034) | -0.001 (0.003) | 0.003 (0.002) |
| observed value | 260 | 260 | 260 | 260 |
| AR(1) | 0.000 | 0.000 | 0.000 | 0.004 |
| AR(2) | 0.639 | 0.605 | 0.843 | 0.509 |
| Hansenp | 0.339 | 0.513 | 0.899 | 0.998 |

Robust standard errors are in parentheses; p-values for each test are reported.

significantly benefits non-durable consumption. This is in line with the findings of the prior study.

After the sample of municipalities was removed, neither the sign nor the significance of the main explanatory variables related to housing support expenditures changed significantly. This suggests that the conclusions are valid even after the robustness regression utilizing adjusted data.

## 5. Discussion

The innovations of this study include the following: first, it explores the consumption effects of different forms of government housing support expenditures. In the analysis, this paper uses China's 1999–2009 affordable housing investment data to represent housing support expenditures in physical form and the 2010–2021 fiscal housing support expenditure data to represent housing support expenditures in monetary form. It is concluded that regardless of the form of housing support expenditures, all of them significantly contribute to the total consumption expenditures of urban residents but have different impacts on the consumption structure. The above conclusion extends the current research of scholars. Secondly, we categorize the consumption structure of urban residents and separately examine the impact of housing support expenditure on residents' consumption structure. The empirical results conclude that housing support expenditures have a significant positive relationship with residents' consumption of essential commodities and favor the consumption of durable goods. Third, subsidies in monetary form have a greater impact on residents' consumption than subsidies in kind. From the empirical results, it can be concluded that all other things being equal, housing support expenditures in monetary form are more likely to promote urban residents' consumption.

This study has strong practical guidance, and the government should correctly understand and grasp the phenomenon and essence of the housing support system, as well as guide urban residents' consumption upgrade in a more targeted way. The study shows that increasing the supply of guaranteed housing and monetary subsidies can promote the total consumption of

urban residents, while monetary subsidies can promote the consumption of non-essential and non-durable commodities of urban residents, indicating that the expenditure on monetary housing support can promote the further release of the consumption potential of urban residents. Under the government's housing support system, the consumption structure and behavior of urban residents are heterogeneous, and their pursuit and realization paths of consumption upgrading should also be different, and there should be a difference in the focus of improving the housing support policy. Urban residents' consumption growth is less constrained by liquidity, and their willingness to spend on non-essential consumption, such as education and entertainment, is higher, so it is possible to satisfy their all-round, deep-level consumption needs by increasing monetary housing subsidies, releasing consumption potential, and promoting consumption upgrading.

This paper also has some shortcomings. The first is that in addition to the core explanatory variables, this paper selects some variables that have a more important influence on urban residents' consumption expenditure as control variables. However, many variables impact urban residents' consumption expenditure, so this paper is slightly weak in this aspect. The second is that this paper utilizes panel data analysis but does not take regional differences into account. Especially in recent years, against the background of the rapid rise of commodity residential prices in some eastern provinces and cities, the impact of housing support expenditures on urban residents' consumption may be different between regions, and this paper does not consider this aspect too much. Of course, this is an inspiration for the authors, and we will take this as a focus of future research.

## 6. Conclusions and policy recommendations

Based on China's provincial panel data, this paper examines the impact of government housing support expenditures on the total consumption and consumption structure of urban residents. The findings show that an increase in government housing support expenditure has a significant contributing effect on the improvement of urban residents' total consumption. The heterogeneity study finds that an increase in the supply of guaranteed housing promotes residents' consumption of essential and durable goods, while an increase in monetary housing guarantee expenditure promotes residents' consumption of non-essential and non-durable goods. As a result, this paper concludes that the increase in housing support expenditure is conducive to enhancing urban residents' total consumption, while the monetary form of housing support expenditure is more conducive to adjusting the consumption structure.

The findings of this paper show that government housing support expenditure significantly increases the total consumption expenditure of urban residents while it has different effects on the structure of residents' consumption expenditure, which is of great practical and theoretical significance for promoting China's construction of a multi-level and long-term housing support system. From the theoretical level, there are more studies on the impact of housing support expenditures on the total consumption expenditures of urban residents but fewer on the impact of the structure of residents' consumption. Moreover, in the existing literature, there are fewer studies comparing and arguing about the in-kind and monetary subsidies of China's housing support from an empirical point of view. In this paper, the consumption effect of housing support expenditure is analyzed at the level of total consumption and consumption structure. It expands on the existing literature to present a more detailed explanation of the mechanism, providing references and lessons for promoting the improvement of the housing support system. From the practical level, this study points out that policymakers should pay attention to the important role of consumption structure while focusing on total consumption. The findings of this paper suggest that when housing support policies are formulated, they

should be considered from the perspective of residents' consumption, categorized and grouped into categories and subgroups of residents' consumption ability, and combined with multi-level classification to build a long-term mechanism to promote consumption enhancement.

## Supporting information

**S1 File.**
(ZIP)

## Author Contributions

**Conceptualization:** Li Shang, Decai Tang.

**Data curation:** Aijun Sun.

**Formal analysis:** Li Shang, Decai Tang, Xiaoling Zhang, Cunshu Li, Nan Pan, Chunfang Huang, Aijun Sun.

**Investigation:** Li Shang.

**Methodology:** Li Shang.

**Project administration:** Li Shang, Decai Tang.

**Resources:** Li Shang.

**Software:** Li Shang.

**Supervision:** Decai Tang, Xiaoling Zhang.

**Validation:** Li Shang, Decai Tang, Xiaoling Zhang.

**Visualization:** Decai Tang, Aijun Sun.

**Writing – original draft:** Li Shang.

**Writing – review & editing:** Li Shang, Decai Tang, Xiaoling Zhang, Cunshu Li, Nan Pan, Chunfang Huang, Aijun Sun.

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
