## [Decision Letter · Decision Letter 0]

22 Feb 2024

PONE-D-23-36493Research on the Economic Effects of Housing Support Expenditures under the Perspective of Consumption Heterogeneity: Evidence from ChinaPLOS ONE

Dear Dr. Tang,

Thank you for submitting your manuscript to PLOS ONE. After careful consideration, we feel that it has merit but does not fully meet PLOS ONE’s publication criteria as it currently stands. Therefore, we invite you to submit a revised version of the manuscript that addresses the points raised during the review process.

We look forward to receiving your revised manuscript.

Kind regards,

Muhammad Khalid Bashir, PhD

Academic Editor

PLOS ONE

Journal Requirements:

Additional Editor Comments:

Since both the reviewers have serious concerns about the presentation of the paper, authors are suggested to go over the suggestions very carefully. 

Reviewers' comments:

Reviewer's Responses to Questions

**Comments to the Author**

1. Is the manuscript technically sound, and do the data support the conclusions?

Reviewer #1: Partly

Reviewer #2: Yes

2. Has the statistical analysis been performed appropriately and rigorously? 

Reviewer #1: N/A

Reviewer #2: Yes

3. Have the authors made all data underlying the findings in their manuscript fully available?

Reviewer #1: No

Reviewer #2: Yes

4. Is the manuscript presented in an intelligible fashion and written in standard English?

Reviewer #1: No

Reviewer #2: Yes

5. Review Comments to the Author

Reviewer #1: Abstract

1.The research contributions of the paper should be articulated more clearly. The abstract is not representative of the content and contributions of the paper. The abstract does not seem to properly convey the rigor of research.

2. Aside from the aim stated in the title, the research gap and the goals of the research are not specified which leads to the reader missing the significance of the research.

3. Some conjunctions have incorrect usage.

4. The abstract is too long and needs to be streamlined.

Introduction

5. The introduction section is detailed, but needs a significant amount of reorganization. It could be strengthened by adding more recent references.

6. Please add as sentence or two to clearly recap how your study differs from what has already been done in literature to ascertain the contributions more strongly

7. More explanation is needed for where there is a research gap and what the goals of the research are. The research gap and the goals of the research are not explained in detail which leads to the reader missing the significance of the research.

8. The research idea should be linked to multiple problems the research is trying to address so that the findings have relevance.

9. The introduction is too long, and it mainly introduces the history of China's housing market rather than the significance of research. It needs to be rewritten.

10. The introduction needs to provide more background information

11. The author needs to clearly point out existing knowledge gaps or conflicting research results in the literature

12. At the end of the introduction, it is best for the author to clearly state the research objectives and the questions that the research aims to answer.

Literature review

13. Section 2.1.1 is not related to the research of the manuscript. I suggest linking this section to the housing market or housing consumption.

14. A research paper is not a graduation thesis, and the author needs to describe the research content of the manuscript clearly, rather than writing all the content. Sections 2.1.1 and 2.1.2 have no connection to the research topic of the manuscript.

15. Sources are out of date. More recent studies (2020-2023) should be included. Also, it should lead up to the research questions in a logical manner.

16. Please discuss learning theories and tie them to both, the research gap addressed by the paper as well as to the factors in the research model.

17.The literature review is insufficient in its addressal of the research gap and research model. It not only needs to be expanded in length, but also in terms of the quality of the content.

18. I don't understand the meaning of section 2.2, is it related to the research content of the manuscript?

19. The content of section 2.3 needs to be reorganized, and the description of the content is very unclear.

20. A suggested approach is to present very specific research questions and then tie the literature review to these questions.

Variable setting

21. Some sentences are too long and very unclear. I suggest the author reorganize the language and change the long sentences to short ones.

22. I suggest changing the descriptive statistics. Mean and standard deviation may not be reliable indicators for some of the non-continuous outcome variables measured. It would be better to add inter-quartile range.

23. The methodology section needs more details and a drastic revision.

24. The reason for using specific analysis is not clearly mentioned. Justification for using a specific methodology or instrument will make it more understandable. Adding more details in this section can give more clarity to the readers.

25. The methodology used should be justified in the article in the light of the research questions (i.e. why is the chosen methodology the best approach to answer the research questions).

Discussion

26. Improve the discussion section to better ascertain what is unique / novel about your findings

27. Explain in detail how the article contributes to new knowledge in the domain.

28. This section needs to be re-written keeping in mind the research questions.

29. The value of the research to the academician and the practitioner should be expressed in an unambiguous manner.

30. The generalizability of the findings must be discussed.

Conclusion

31. Update the conclusion to include the newly formulated theoretical contributions.

32. Summarize the key results in a compact form and re-emphasize their significance.

33. Summarize how the article contributes to new knowledge in the domain.

Reviewer #2: This study empirically investigates how government housing support spending affects urban families' overall and consumption-structured consumption using panel data at the province level.

The salient features of the current study are:

• The significance of the study is intelligibly highlighted

• The research problem is genuine and has regional and local appeal for welfare outcomes

• The research gap is well-identified and bridged

• The theoretical framework is constructed on simple but sound and pertinent theories of economics

• Operational definitions of the variables used in the model are very clear and their measurements (and units of measurement) are appropriate

• The results are logically interpreted and discussed in comparison with the previously conducted relevant studies

To further improve the quality of the manuscript, the following suggestions are proposed:

1. The abstract is too long, especially, the part of the abstract presenting the study's findings. So, it must be shortened. The last two sentences concluding the abstract are confusing and, hence, need to be clarified.

2. On page 4 (lines 161-162) clarify the sentence: “Beneficial increase in local residents' economic activity [26]”.

3. On page 6 (lines 242-253): use the punctuation and that of the sentence case appropriately.

4. On page 10 (lines 383-384): clarify the sentence: “This paper will be removed from the sample of the Shanghai Municipality, Tibet Autonomous 384 Region, as more data is missing.”

5. Justify, in tables 7, 8, and 9, for GMM estimation, why and how each explained variable is split into two.

6. PLOS authors have the option to publish the peer review history of their article (what does this mean?). If published, this will include your full peer review and any attached files.

Reviewer #1: No

Reviewer #2: No

---

## [Author Response · Author response to Decision Letter 0]

28 Mar 2024

Response to Reviewer 1

Abstract

Q1: The research contributions of the paper should be articulated more clearly. The abstract is not representative of the content and contributions of the paper. The abstract does not seem to properly convey the rigor of research.

Q2: Aside from the aim stated in the title, the research gap and the goals of the research are not specified which leads to the reader missing the significance of the research.

Q3: Some conjunctions have incorrect usage.

Q4: The abstract is too long and needs to be streamlined.

Response: Thank you, reviewers, for these insightful and educative comments. We believe Q1-Q4 are related and have been revised accordingly in the manuscript. It will be quite difficult to provide answers for each of them due to their arrangements in the revised manuscript. However, all comments have been addressed in detail in the abstract section. Please see Lines 9-21 in the abstract of the revised manuscript for your reference.

Introduction

Q5: The introduction section is detailed, but needs a significant amount of reorganization. It could be strengthened by adding more recent references.

Response: Thank you for your advice. We have re-written the introduction and added new references; Please see Lines 25-71 in the introduction of the revised manuscript for your reference.

Q6: Please add as sentence or two to clearly recap how your study differs from what has already been done in literature to ascertain the contributions more strongly.

Response: Thank you, reviewer. In the introduction, we have identified how this study complements and extends existing research. Please see lines 157-161 of the manuscript for detailed revisions.

Q7: More explanation is needed for where there is a research gap and what the goals of the research are. The research gap and the goals of the research are not explained in detail which leads to the reader missing the significance of the research.

Q8: The research idea should be linked to multiple problems the research is trying to address so that the findings have relevance.

Q9: The introduction is too long, and it mainly introduces the history of China's housing market rather than the significance of research. It needs to be rewritten.

Q10: The introduction needs to provide more background information

Q11: The author needs to clearly point out existing knowledge gaps or conflicting research results in the literature

Q12: At the end of the introduction, it is best for the author to clearly state the research objectives and the questions that the research aims to answer.

Response: Thank you for your instructive comments. We believe that questions 7-12 are related and cannot be answered individually; therefore, they are presented here in a unified manner. We have carefully studied your comments and made the following major changes to the introduction: first, we have rewritten the introduction; second, in the new introduction, we have highlighted the research questions, research objectives, and significance of the study; and lastly, we have deleted the original redundancy and streamlined the content. Please refer to the introduction section of this paper for detailed revisions.Please see Lines 25-71 in the introduction of the revised manuscript for your reference.

Literature review

Q13: Section 2.1.1 is not related to the research of the manuscript. I suggest linking this section to the housing market or housing consumption.

Response: Thank you for the suggestions. We have deleted this section from the original manuscript and rewritten the chapter from the perspective of housing and human consumption, as detailed Lines 75-106 in section 2.1 of the new manuscript.

Q14: A research paper is not a graduation thesis, and the author needs to describe the research content of the manuscript clearly, rather than writing all the content. Sections 2.1.1 and 2.1.2 have no connection to the research topic of the manuscript.

Response: Thank you for your advice. We have removed such sections.

Q15: Sources are out of date. More recent studies (2020-2023) should be included. Also, it should lead up to the research questions in a logical manner.

Response: Thank you for your comments. We have updated the literature, most of which is dated 2020-2023. Please see Lines 74-135 of the new manuscript for detailed revisions.

Q16: Please discuss learning theories and tie them to both, the research gap addressed by the paper as well as to the factors in the research model.

Response: Thank you very much for your advice. We have reorganized the theoretical part and analyzed it in conjunction with the relevant variables. Please see Lines 137-171 of the revised manuscript for details.

Q17: The literature review is insufficient in its addressal of the research gap and research model. It not only needs to be expanded in length, but also in terms of the quality of the content.

Response: Thank you for your comments. We have reorganized and rewritten the literature review section. Detailed revisions can be found in the Lines 75-135 of the new manuscript.

Q18: I don't understand the meaning of section 2.2, is it related to the research content of the manuscript?

Response: Thank you for the question. We have removed that section from the manuscript.

Q19: The content of section 2.3 needs to be reorganized, and the description of the content is very unclear.

Response: Thank you for your comments on this section. We have reorganized and rewritten that section in detail. Please see Lines 137-171 of the new manuscript for detailed revisions.

Q20: A suggested approach is to present very specific research questions and then tie the literature review to these questions.

Response: Thank you for your valuable comments. We have already set out the specific questions to be examined in this paper in the latter part of Chapter 2 ( Lines 158-161 ), taking into account the research of other scholars. Detailed revisions can be found in lines 158-161 of the revised draft.

Variable setting

Q21: Some sentences are too long and very unclear. I suggest the author reorganize the language and change the long sentences to short ones.

Response: Thank you for your comments. We have reworked the linguistic part of the manuscript, changing long sentences into short sentences. Please see the new manuscript for detailed revisions.

Q22: I suggest changing the descriptive statistics. Mean and standard deviation may not be reliable indicators for some of the non-continuous outcome variables measured. It would be better to add inter-quartile range.

Response: Thanks for the suggestion. We have added an inter-quartile range to the descriptive statistics. Detailed results are shown in Tables 2 and 3 in the text.

Q23: The methodology section needs more details and a drastic revision.

Q24: The reason for using specific analysis is not clearly mentioned. Justification for using a specific methodology or instrument will make it more understandable. Adding more details in this section can give more clarity to the readers.

Q25: The methodology used should be justified in the article in the light of the research questions (i.e. why is the chosen methodology the best approach to answer the research questions).

Response: Thank you very much for your advice on the methods section. In response to questions 23, 24, and 25, we have made the following changes: first, a brief description of the chosen methodology; second, a detailed justification for selecting the corresponding methodology. The specific changes can be found in lines 250-256 and 295-306 of the manuscript.

Discussion

Q26: Improve the discussion section to better ascertain what is unique / novel about your findings

Response: Thank you, reviewer, for this insight. We have revised the discussion section to highlight the innovative aspects of the findings. Please see Lines 402-434 in the revised manuscript for your reference.

Q27: Explain in detail how the article contributes to new knowledge in the domain.

Q28: This section needs to be re-written keeping in mind the research questions.

Q29: The value of the research to the academician and the practitioner should be expressed in an unambiguous manner.

Response: We appreciate the insightful and educative comments. We consider questions 27-29 relevant and have amended the above. The value and contribution of this paper have been explained and expressed based on a write-up of the discussion section. Please see Lines 402-434 for the revisions.

Q30: The generalizability of the findings must be discussed.

Response: Thank you for your suggestion. In the final section of the Discussion section, the limitations and generalizability of the study's findings are discussed. Please see lines 415-426 for detailed revisions.

Conclusion

Q31: Update the conclusion to include the newly formulated theoretical contributions.

Q32: Summarize the key results in a compact form and re-emphasize their significance.

Q33: Summarize how the article contributes to new knowledge in the domain.

Response: Thank you, reviewers, for your valuable comments. We believe that questions 31-33 are closely related and are answered here in a unified manner. We have revised the conclusion section based on the comments, re-written the conclusion section, and made the following corrections: first, we have abbreviated the research results section of this paper to make it more compact and precise; second, we have focused on emphasizing the theoretical contributions and practical significance of this paper. Detailed revisions can be found in Lines 437-459 of the manuscript.

Response to Reviewer 2

 This study empirically investigates how government housing support spending affects urban families' overall and consumption-structured consumption using panel data at the province level.

The salient features of the current study are:

• The significance of the study is intelligibly highlighted

• The research problem is genuine and has regional and local appeal for welfare outcomes

• The research gap is well-identified and bridged

• The theoretical framework is constructed on simple but sound and pertinent theories of economics

• Operational definitions of the variables used in the model are very clear and their measurements (and units of measurement) are appropriate

• The results are logically interpreted and discussed in comparison with the previously conducted relevant studies.

Response: Thank you, reviewers, for seeing merit in our work.

Q1: The abstract is too long, especially, the part of the abstract presenting the study's findings. So, it must be shortened.

Q2: The last two sentences concluding the abstract are confusing and, hence, need to be clarified.

Response: Thank you for these insightful comments. We have revised the concerns you raised in the manuscript accordingly. Details of the revisions can be found in Lines 9-21 of the paper, where we have reorganized and re-written it.

Q3: On page 4 (lines 161-162) clarify the sentence: “Beneficial increase in local residents' economic activity [26]”.

Response: Thank you for your careful review. We have reorganized and rewritten the literature review. Please see Lines75-135 the revised manuscript for detailed revisions.

Q4: On page 6 (lines 242-253 ): use the punctuation and that of the sentence case appropriately.

Response: Thank you. We have reorganized and rewritten it. Please refer to section 2.2 ( Lines 137-171 ) of the text for detailed revisions.

Q5: On page 10 (lines 383-384): clarify the sentence: “This paper will be removed from the sample of the Shanghai Municipality, Tibet Autonomous 384 Region, as more data is missing.”

Response: Thank you for the suggestion. We have clarified this section. Please see Lines 180-186 of the new paper for more details.

Q6: Justify, in tables 7, 8, and 9, for GMM estimation, why and how each explained variable is split into two.

Response: Tables 7-9 explore the consumption structure of urban residents. Separate bases, relevance to housing, necessity, and durability. This paper divides the structure of consumption expenditures into three aspects: housing-related consumption expenditures (variable Consume1 in Table 7), housing-unrelated consumption expenditures (variable Consume2 in Table 7); necessity consumption expenditures (variable Consume3 in Table 8), non-necessity consumption expenditures (variable Consume4 in Table 8); and consumption expenditures on durable goods (variable Consume5 in Table 9), and consumption expenditures on non-durable goods (variable Consume6 in Table 9).

---

## [Decision Letter · Decision Letter 1]

11 Jun 2024

Research on the Economic Effects of Housing Support Expenditures under the Perspective of Consumption Heterogeneity: Evidence from China

PONE-D-23-36493R1

Dear Dr. Tang,

We’re pleased to inform you that your manuscript has been judged scientifically suitable for publication and will be formally accepted for publication once it meets all outstanding technical requirements.

Kind regards,

Muhammad Khalid Bashir, PhD

Academic Editor

PLOS ONE

Additional Editor Comments (optional):

The authors need to address the minor comments / suggestions of reviewer 1

Reviewers' comments:

Reviewer's Responses to Questions

**Comments to the Author**

1. If the authors have adequately addressed your comments raised in a previous round of review and you feel that this manuscript is now acceptable for publication, you may indicate that here to bypass the “Comments to the Author” section, enter your conflict of interest statement in the “Confidential to Editor” section, and submit your "Accept" recommendation.

Reviewer #1: All comments have been addressed

Reviewer #2: All comments have been addressed

2. Is the manuscript technically sound, and do the data support the conclusions?

Reviewer #1: Partly

Reviewer #2: Yes

3. Has the statistical analysis been performed appropriately and rigorously? 

Reviewer #1: No

Reviewer #2: Yes

4. Have the authors made all data underlying the findings in their manuscript fully available?

Reviewer #1: Yes

Reviewer #2: Yes

5. Is the manuscript presented in an intelligible fashion and written in standard English?

Reviewer #1: Yes

Reviewer #2: Yes

6. Review Comments to the Author

Reviewer #1: * Is the data on the construction of guaranteed housing construction in China so outdated? Don't you have the latest data?

* Line 24-28, what do these sentences mean?

* Why use logarithmic data? Tables 2 and 3 require raw data.

Reviewer #2: The authors have revised the manuscript according to the concerns raised.

Details of the revisions can be found in Lines 9-21, 75-135, 137-171, 180-186, and Tables 7-9, where they have reorganized, re-written, and clarified.

7. PLOS authors have the option to publish the peer review history of their article (what does this mean?). If published, this will include your full peer review and any attached files.

Reviewer #1: No

Reviewer #2: **Yes: **Tusawar Iftikhar Ahmad

---

## [Editor Report · Acceptance letter]

21 Jun 2024

PONE-D-23-36493R1 

PLOS ONE

Dear Dr. Tang, 

I'm pleased to inform you that your manuscript has been deemed suitable for publication in PLOS ONE. Congratulations! Your manuscript is now being handed over to our production team.

Kind regards, 

on behalf of

Dr. Muhammad Khalid Bashir 

Academic Editor

PLOS ONE